# A Synchrotron Mössbauer Spectroscopy Study of a Hydrated Iron-Sulfate at High Pressures

**Tyler Perez \*,†, Gregory J. Finkelstein ††, Olivia Pardo, Natalia V. Solomatova ††† and Jennifer M. Jackson \***

Seismological Laboratory, California Institute of Technology, Pasadena, CA 91125, USA; gjfinkel@princeton.edu (G.J.F.); opardo@caltech.edu (O.P.); natalia.solomatova@ens-lyon.fr (N.V.S.)

* Correspondence: tperez11@jhu.edu (T.P.); jackson@gps.caltech.edu (J.M.J.)

† Now at Department of Earth and Planetary Sciences, Johns Hopkins University, Baltimore, MD 21218, USA.

†† Now at Department of Geosciences, Princeton University, Princeton, NJ 08544, USA.

††† Now at École Normale Supérieure de Lyon, Laboratoire de Géologie, CNRS, UMR 5276, 69342 Lyon, France.

**Abstract:** Szomolnokite is a monohydrated ferrous iron sulfate mineral, $FeSO_4 \cdot H_2O$, where the ferrous iron atoms are in octahedral coordination with four corners shared with $SO_4$ and two with $H_2O$ groups. While somewhat rare on Earth, szomolnokite has been detected on the surface of Mars along with several other hydrated sulfates and is suggested to occur near the surface of Venus. Previous measurements have characterized the local environment of the iron atoms in szomolnokite using Mössbauer spectroscopy at a range of temperatures and 1 bar. Our study represents a step towards understanding the electronic environment of iron in szomolnokite under compression at 300 K. Using a hydrostatic helium pressure-transmitting medium, we explored the pressure dependence of iron's site-specific behavior in a synthetic szomolnokite powdered sample up to 95 GPa with time-domain synchrotron Mössbauer spectroscopy. At 1 bar, the Mössbauer spectrum is well described by two $Fe^{2+}$-like sites and no ferric iron, consistent with select conventional Mössbauer spectra evaluations. At pressures below 19 GPa, steep gradients in the hyperfine parameters are most likely due to a structural phase transition. At 19 GPa, a fourth site is required to explain the time spectrum with increasing fractions of a low quadrupole splitting site, which could indicate the onset of another transition. Above 19 GPa we present three different models, including those with a high- to low-spin transition, that provide reasonable scenarios of electronic environment changes of the iron in szomolnokite with pressure. We summarize the complex range of $Fe^{2+}$ spin transition characteristics at high-pressures by comparing szomolnokite with previous studies on ferrous-iron bearing phases.

**Keywords:** iron-sulfates; hydrous phases; szomolnokite; high-pressure; synchrotron Mössbauer spectroscopy; spin-transition

## 1. Introduction

Historically, sulfate minerals have been studied mainly in the context of surface processes such as evaporitic deposits and hydrothermal systems [1] or in the context of mine tailings and wastes [2]. Sulfate salts play important roles in the cycling of metals and sulfates in terrestrial systems [3], and are thought to play important roles on the surface weathering processes on other planetary bodies. In particular, relatively large deposits of monohydrated sulfates have been detected on the surface of Mars using absorption spectroscopy [4,5], with Lichtenberg et al. [5] specifically preferring szomolnokite to explain certain signatures present in Aram Chaos. Talla and Wildner [6] perform a detailed

spectroscopic study on the kieserite-szomolnokite solid-solution series under ambient and Martian temperature conditions, due to the high probability that intermediate compositions along this join exist on Mars. Additionally, Lane et al. [7] suggests that hydrous iron sulfates closely match Mini-TES (Miniature Thermal Emission Spectrometer) and Mössbauer data from Martian soil analyzed by MER (Mars Exploration Rovers). Chou et al. [3] performed experiments on the stability of a variety of hydrated and anhydrous sulfates in Martian reaction environments. They suggest that hydrated sulfates could play important roles in the hydrologic cycle of Mars. There is also evidence for hydrated sulfates on the surface of Europa [8]**,** and they have been investigated up to pressures of ~2.5 GPa as possible constituents of icy moon mantles [9]. Barsukov et al. [10] suggested that barium and strontium sulfates could possibly be stable in the crust in Venus, although it is not clear if they would be formed sub-surface as a result of dehydration processes.

Previous experimental studies on hydrated sulfates at elevated pressures include Fortes et al. [11] who used neutron powder diffraction of deuterated $MgSO_4 \cdot 11D_2O$ to explore phase transitions within the range $0.1 < P < 1000$ MPa and 150 to 280 K. $MgSO_4 \cdot 11D_2O$ is the deuterated analogue of meridianiite, which is triclinic with space group $P\overline{1}$ (Z = 2). They detected evidence of peritectic melting at 0.545 GPa and 275 K as well as a phase transition at 0.9 GPa and 240 K, decomposing into ice VI + $MgSO_4 \cdot 9D_2O$ which is monoclinic with space group $P2_1/c$ (Z = 4). Previous Mössbauer work on szomolnokite ($FeSO_4 \cdot H_2O$) has focused largely on the effect of temperature on the hyperfine parameters at ambient pressure. In a recent comprehensive review paper, Dyar et al. [12] presented and summarized conventional Mössbauer spectra evaluations for a suite of iron-bearing sulfates. Alboom et al. [13] studied szomolnokite with energy domain Mössbauer spectroscopy from 4.2 K to 450 K and found a magnetic order-disorder transition at 29.6 ± 0.5 K. Giester et al. [14] used Mössbauer spectroscopy as well as X-ray diffraction to find that for $(Fe,Cu)SO_4 \cdot H_2O$, there is a reduction in symmetry from monoclinic to triclinic beyond 20 mol % Cu and report a magnetic order transition between 15 K and 4.2 K at ambient pressure.

As discussed in Meusburger et al. [15], szomolnokite ($FeSO_4 \cdot H_2O$) is isostructural to kieserite ($MgSO_4 \cdot H_2O$), which is monoclinic and in space-group $C2/c$. The kieserite structure consists of corner sharing $[MO_4(H_2O)]^{6-}$ units, which run parallel to the crystallographic $c$-axis. Lattice parameters for szomolnokite obtained in this study and previous studies [14–17] are discussed below and given in Table S1. Meusberger et al. [15] studied the structural evolution of szomolnokite up to pressures of 9.2 GPa by means of X-ray diffraction, Fourier-transform infrared spectroscopy, and Raman spectroscopy. They found a transition from monoclinic phase in space group $C2/c$ to triclinic phase in space group $P\overline{1}$. They find that the transition is ferroelastic and second order in thermodynamic character.

A recent study has characterized the magnetic and structural changes of jarosite, a hydroxylated iron sulfate, $KFe_3(OH)_6(SO_4)_2$, as a function of pressure (up to 40 GPa) using a variety of techniques, including synchrotron Mössbauer spectroscopy [18]. They described the iron environment with a single $Fe^{3+}$ site which steadily increases in quadrupole splitting and steadily decreases in isomer shift as pressure increases. They found a continuous pressure induced phase transition that did not affect the spin state of the $Fe^{3+}$. They also reported a dramatic increase in magnetic ordering temperature as a function of pressure; up to 240 K at 40 GPa.

In this study, we use time-domain synchrotron Mössbauer spectroscopy to examine the high-pressure behavior of szomolnokite at the crystal chemical level. Mössbauer spectroscopy directly probes the local electronic environment of iron atoms within solids, which can provide valuable physical insight to transitions occurring in the examined phase under high pressures. Understanding the high-pressure behavior of szomolnokite will provide a step towards characterizing the behavior of complex hydrated minerals under compression and has implications for planetary interiors.

## 2. Materials and Methods

The szomolnokite powder used in this study was synthesized through a collaboration with Isoflex ($FeSO_4 \cdot H_2O$, using 96% $^{57}Fe$). For ambient pressure X-ray diffraction characterization, a powdered sample of szomolnokite contained in a Kapton tube was used. X-ray diffraction measurements were taken at beamline 12.2.2 at the Advanced Light Source (ALS) operating with a wavelength of 0.4972 Å and

beam size of 20 μm² (full width at half maximum). The integrated ambient pressure X-ray diffraction pattern is shown in Figure 1 and was fit using Rietveld refinement in GSAS-II [19]. The resulting lattice parameters are in agreement with previously reported values [14–17] (Table S1).

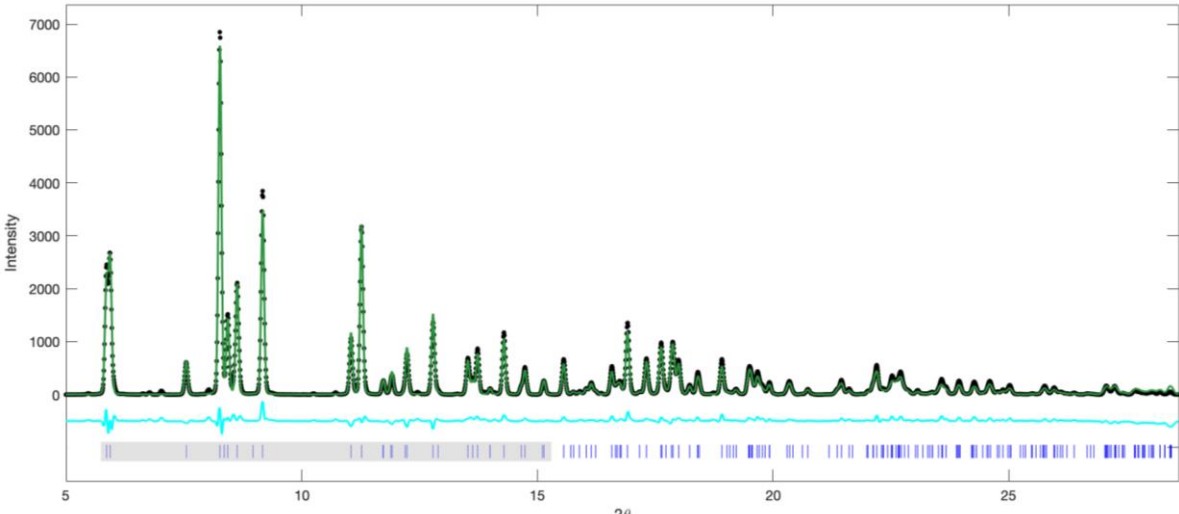

**Figure 1.** Integrated ambient pressure-temperature diffraction pattern with predicted reflections for szomolnokite. The background-corrected integrated pattern (black dots) is fit using Rietveld refinement in GSAS-II. The resulting fit is shown in green with residuals below (cyan). Predicted reflections for the monoclinic *C*2/*c* phase are shown in blue. The reflections highlighted in the grey region are chosen to calculate the ambient pressure lattice parameters: a = 7.09 Å, b = 7.55 Å, c = 7.78 Å, β = 118.65(2)°, volume = 365.2(1) Å³. Comparison to previous studies are shown in Table S1.

For the synchrotron Mössbauer spectroscopy measurements, a powdered $^{57}FeSO_4 \cdot H_2O$ sample (<5 μm thick) was loaded into a Princeton-design symmetric diamond anvil cell, assembled with two one-quarter-carat diamonds with 250 μm diameter culets, plus 50 μm bevels, mounted on tungsten-carbide seats. A rhenium gasket was indented to ~50 μm and drilled with a 130-μm hole, which was then loaded with the powdered szomolnokite sample. Two rubies were placed in the sample chamber to be used as pressure gauges. The cell was placed into a vacuum of ~$10^{-5}$ Pa to remove any residual liquids before it was loaded with helium under ~0.17 GPa (25,000 pounds per square inch).

The hyperfine interactions, namely the quadrupole splitting (QS) and isomer shift (IS), were determined using time-domain synchrotron Mössbauer spectroscopy (SMS), a technique that probes the local electronic environment of $^{57}Fe$ atoms in the sample. The isomer shift is proportional to the s-electron density at the nucleus while the quadrupole splitting describes the asymmetry in the electric field gradient at the Mössbauer nucleus. Knowledge of both quantities provides constraints on the valence and spin state of the iron atoms in szomolnokite. The distribution of quadrupole splittings for a particular Mössbauer site (expressed as the full width at half maximum, FWHM, in units of mm/s) as well as site weight fraction was also determined in the fitting. The SMS measurements were conducted at Sector 3-ID-B of the Advanced Photon Source of Argonne National Laboratory. The storage ring was operated in 24 bunch mode, top-up, with 153 ns bunch separation. The focus size of the X-ray beam was about 15 × 15 μm². The time window used to evaluate the spectra was 23 ns to 129 ns after excitation. The pressure in the sample chamber was determined before and after the SMS measurement using the ruby fluorescence method [20] and the standard deviation between two ruby spheres proximal to the sample.

## 3. Results

The SMS spectra were fitted with version 2.2.0 of the CONUSS software [21], which uses a least-square algorithm to fit iron's hyperfine parameters in szomolnokite, as well as material properties such as effective thickness and the Lamb–Mössbauer factor. We estimated the sample thickness

before compression to be <5 μm thick, using a calibrated binocular microscope. Although it is known that the Lamb–Mössbauer factor of solids generally increases with increasing pressure, there are no quantitative constraints for szomolnokite. In order to keep the number of fitting parameters at a minimum, the Lamb–Mössbauer factor was fixed at 0.6, and physical thickness was fixed at 3 μm at all compression points. The quadrupole splitting value for each site was fitted. In some cases, the distribution of quadrupole splittings for a particular Mössbauer site (expressed as the full width at half maximum, FWHM, in units of mm/s) could be determined in the fitting, whereas in some models incorporating additional sites, the FWHM was fixed. At 1 bar, the isomer shifts of sites 1 and 2 were determined in a CONUSS fitting procedure called, "dual fit", where the spectrum of the sample and the spectrum of the sample with a 10-μm thick stainless steel reference foil containing the natural $^{57}$Fe enrichment level are fit simultaneously (Tables S2–S4). The best fit results at 1 bar agree reasonably well with the model from Dyar et al. [12], given that the highest weight fraction site falls in the same area in IS vs. QS space as the other szomolnokite samples (Figure 2; Table S5). The primary site also agrees with other ambient pressure Mössbauer studies on szomolnokite. However, the isomer shift of the second site (representing 9% of total iron content) required to fit the spectrum, is relatively low (~1 mm/s) compared with that of previously characterized szomolnokite samples. It is clear, though, that this second site is not associated with ferric iron, as its QS value is relatively high (2.07 $\pm$ 0.03 mm/s), and therefore, we have ruled out the possibility of any $Fe^{3+}$ in our sample. We have also ruled out amorphous components as well as significant texture effects.

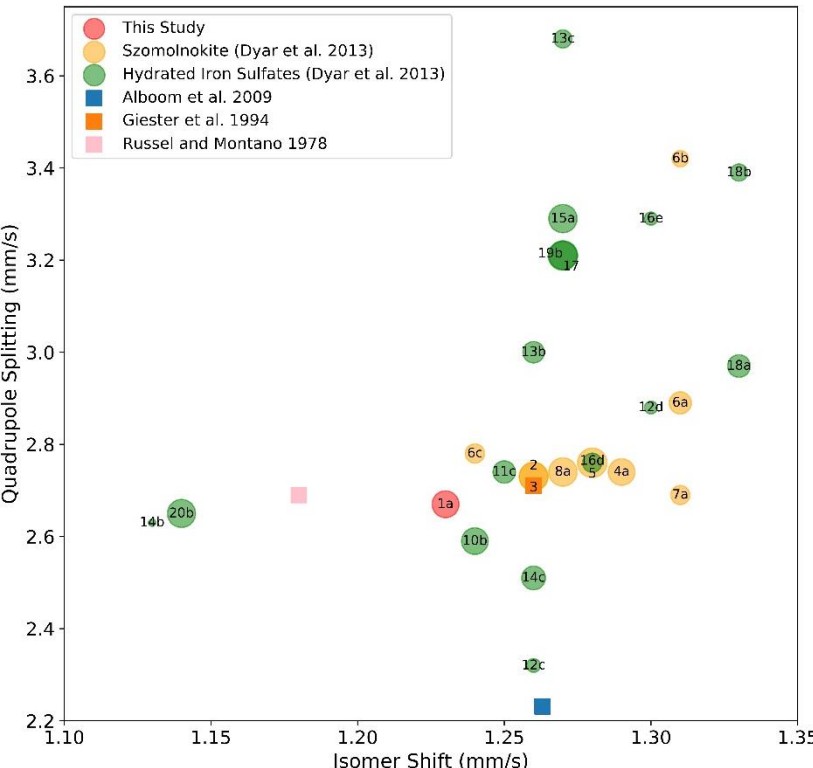

**Figure 2.** Isomer shift relative to bcc-iron versus quadrupole splitting at 0 GPa of $^{57}$Fe-Mössbauer sites in a variety of hydrated iron sulfates. Circles with the same number represent different sites within a single sample. Labels come from [12–14,22] (see Table S5). Size of circle represents weight fraction of that site. The colored squares represent values from earlier Mössbauer studies of szomolnokite. The IS of site 1b from this study is out of the range of this figure at 0.89 $\pm$ 0.02 mm/s.

Although there is only one crystallographic site for iron in szomolnokite at ambient conditions [13–15], two Mössbauer sites are required to fit the data. As discussed in previous works on iron-bearing phases characterized by a single crystallographic site for ferrous iron, such as bridgmanite and (Mg,Fe)O ferropericlase [23–26], additional $Fe^{2+}$-like sites in the data evaluation of Mössbauer

spectra may arise from differences in the next nearest-neighbor environments. Atomic-specific probes like Mössbauer spectroscopy are therefore capable of resolving such local-environment characteristics. At higher pressures, three or more sites are required to fit the spectra. This is not unreasonable if we look to a recent combined single-crystal X-ray diffraction study of szomolnokite [15], where they found two distinct crystallographic sites for iron in the triclinic structure at 7.3 GPa. A recent combined X-ray diffraction and SMS study on $Fe_2Si_2O_6$ ferrosilite also showed that additional iron sites are required to fit the SMS spectra. Specifically, at the structural phase transition of ferrosilite from *C*2/*c* to HP-*P*2$_1$/*c*, Solomatova et al. [27] found that four high-spin Mössbauer sites are required to fit the SMS spectra (M1a, M1b, M2a, M2b), although only two distinct crystallographic sites for iron could be resolved with single-crystal X-ray diffraction (M1 and M2).

The isomer shifts could not be determined as a function of pressure relative to stainless steel foil due to the fact that the sample was smaller than the X-ray focus size, leading to complications in interpreting the scattering effects around its edges with respect to the much larger stainless steel reference foil. We therefore did not attempt to interpret the isomer shift values from the reference foil with compression. Rather, we report the relative difference in isomer shift of the sites with respect to site 1.

The results indicate significant changes in the hyperfine parameters with increasing pressure. We explore three different best-fitting models over the compression range of our study. Although these models produce the same hyperfine parameters for the iron sites at low pressures, the models diverge at pressures above 19 GPa, as discussed below (Figures 3, 4, and S1, and Tables S2–S4).

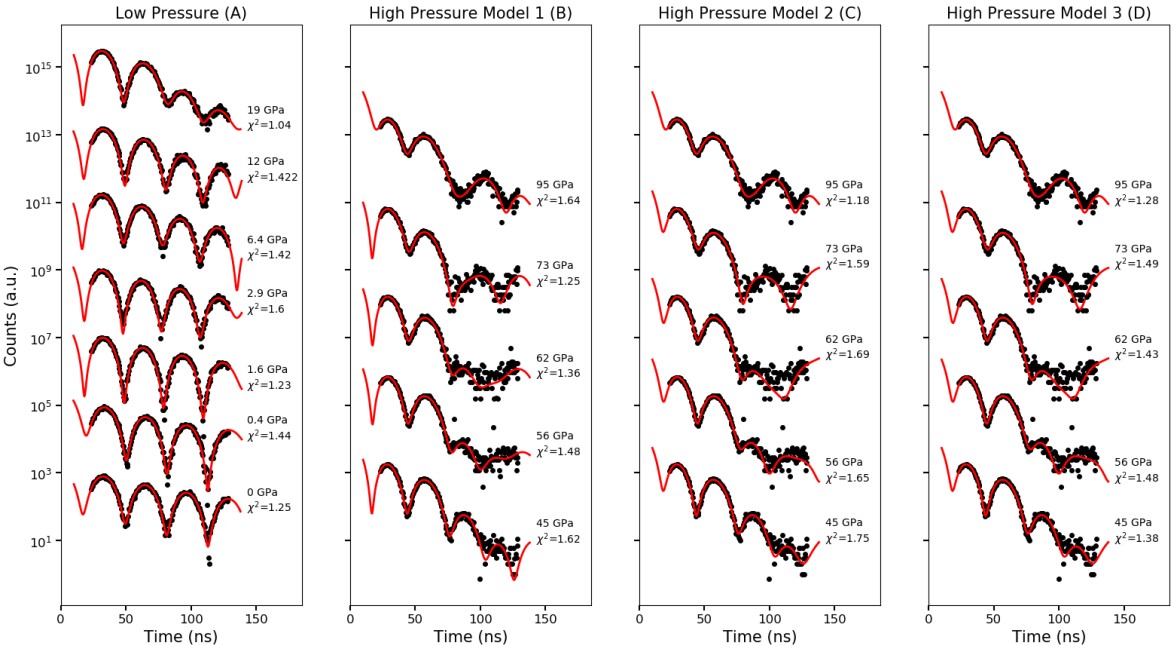

**Figure 3.** Representative synchrotron Mössbauer spectroscopy (SMS) spectra of szomolnokite: black circles are binned data and red curves are best fit results from CONUSS. (**A**) All models are identical from 0 to 19 GPa. (**B**) Model 1 features only 3 sites and assumes all the iron in the material remains in the high-spin state up to 95 GPa. (**C**) Model 2 applied to 19 to 95 GPa. This features an additional site, which could indicate a transition (increased distortion, structural, or high- to low-spin transition) that occurs between 19 and 45 GPa. (**D**) Model 3 applied to the higher pressures. This model features a gradual increase in presence of a low spin site from 19 to 95 GPa.

Three Mössbauer sites were needed to fit the data in the 0.4 to 19 GPa compression range. In this pressure range, the quadrupole splitting fluctuates in each of the three sites (Figure 4). The weight fraction of the primary site rapidly decreases in this pressure range and the other two sites gain that weight fraction. There is a significant change at 19 GPa where the QS for site 3 increases to 2.7 mm/s, and the primary site weight fraction decreases. The isomer shift values relative to site 1 show large

fluctuations in this compression range. The relative IS of site 2 increases sharply before dropping off slightly at 19 GPa (Figure S1). The relative IS of site 3 decreases at low compressions, then increases with pressure up to 19 GPa. These changes are likely related to a structural transition, as discussed in the next section.

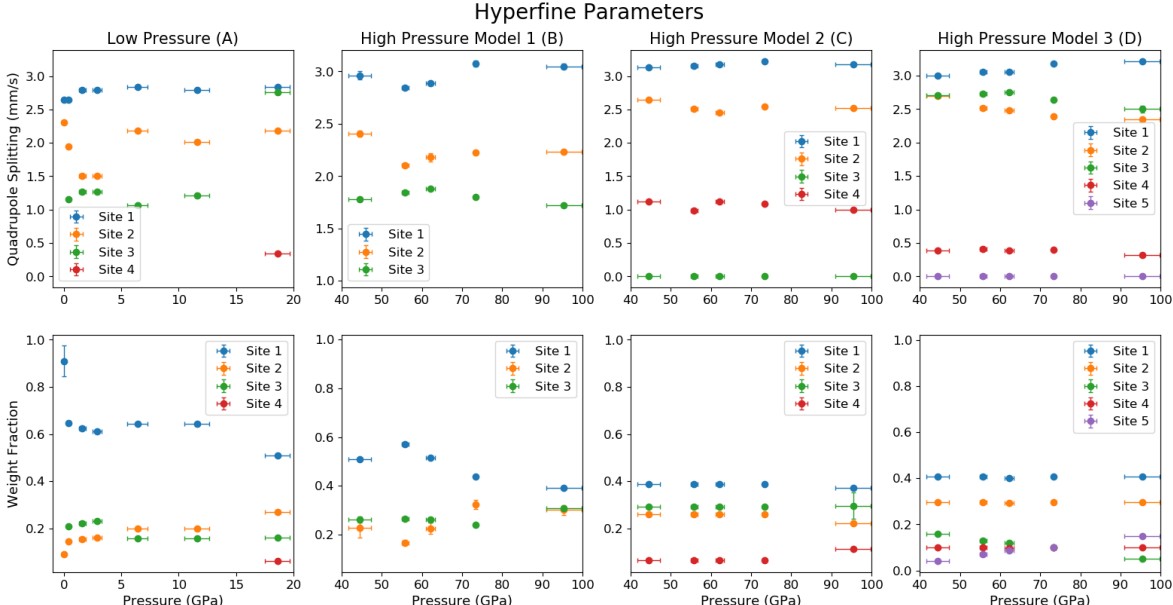

**Figure 4.** Fitted hyperfine parameters of szomolnokite as a function of pressure: quadrupole splitting and weight fractions of the iron sites. (**A**) Low pressure values, as all models are identical from 0 to 19 GPa. (**B**) Model 1 features only 3 sites from 45 to 95 GPa and assumes all the iron in the material remains in the high-spin state. (**C**) Values from Model 2, featuring a high- to low-spin transition in site 3 (quadrupole-splitting of 0 mm/s) between 19 and 45 GPa. (**D**) Values from Model 3 consider a gradual high-spin (site 3) to low-spin (site 5) transition from 45 to 95 GPa: site 3 (green) and site 5 (purple) trade off in weight fraction until site 3 has zero weight fraction and site 5 reaches a weight fraction around 0.2. Values are reported in Tables S2–S4.

Above 19 GPa, a fourth site with a low weight fraction and low quadrupole splitting was introduced to increase the quality of the fit. This site has a small but non-zero quadrupole splitting suggesting that this additional feature could be indicative of another transition. Therefore, above 19 GPa we consider three models that incorporate different transition scenarios, including a high- to low-spin-transition. Model 1 does not incorporate a low spin site (zero quadrupole-splitting) over the compression range of this study and removes the fourth site at 19 GPa. Model 1 shows a small decrease in the QS of sites 1 and 2 above 45 GPa, followed by a steady rise over the remaining pressure range, whereas the QS of site 3 is essentially invariant with compression. With fewer total parameters in Model 1 than the other models, the weight fractions of the sites could freely vary in the fitting procedure without running into correlation problems. We observe a small increase in weight percent of site 1 above 45 GPa followed by a steady decrease with pressure. The weight fraction of site 2 exhibits the opposite trend as site 1, while the weight percent of site 3 shows minimal fluctuations in this pressure range with a small increase at 95 GPa.

Model 2 contains four sites between 19 and 45 GPa, where site 4 has a low QS at 19 GPa (6% of the total iron) and steadily increases with increasing pressure and site 3 has zero QS at 45 GPa (low-spin ferrous iron, 30% of total iron) (Figure 4). This model would reflect a structural transition around 19 GPa and/or a high- to low-spin transition at 45 GPa. Above 45 GPa, the QS and weight fraction remain largely pressure-invariant in this model, except for slight increase in the weight fraction of site 4 at 95 GPa.

Model 3 introduces a fifth Mössbauer site as a low-spin site (zero quadrupole-splitting), which gradually increases in weight fraction (starting at ~4% of the total iron) at the expense of site 3 (high-

spin in this model) between 45 GPa to 95 GPa. This model represents a gradual high- to low-spin transition beginning around 45 GPa. In this model, the QS of site 1 has a weak positive correlation with pressure, whereas the QS of sites 2 and 3 have a weak negative correlation with pressure; the QS of site 4 is independent of pressure within uncertainties.

The relative IS values and broadening (expressed as the FWHM of the QS in mm/s) for the iron sites in these models are plotted as a function of pressure in Figure S1. The relative IS of Model 1 shows sites 2 and 3 decreasing with pressure. In Model 2, the relative isomer shifts are fairly stable with respect to pressure, whereas in Model 3, the IS of site 4 decreases with increasing pressure up to 73.4 GPa. All models feature significant broadening of sites 1, 2, and 3 above 45 GPa (FWHM values range from around 0.3 to 0.6 mm/s) (Figure S1).

## 4. Discussion and Implications

The steep gradient in hyperfine parameters at low pressures (Figure 4; Tables S2–S4) suggests a structural change, as demonstrated by recent single-crystal X-ray diffraction results by Meusberger et al. [15]. At higher pressures, the three different scenarios fit the data well, as illustrated in Figure 3, resulting in an average reduced $\chi^2$ of 1.39 for Model 1 (which does not incorporate a spin transition), 1.44 for Model 2 (sharp spin transition in one site at 45 GPa), and 1.37 for Model 3 (gradual spin transition from 45 to 95 GPa). We find that, while some sites show decreasing QS values, the QS values for several sites across the models increase with increasing pressure. This trend is distinct from the trend highlighted at room-pressure comparing a range of iron-bearing minerals [12], which found that the quadrupole splitting tends to have a positive correlation with bond length. An increase in the QS values with pressure can be correlated with the combination of bond-shortening and other complex changes influencing the local electronic environments crystal structure, including lattice distortion [23]. This suggests an increasingly distorted (non-cubic) local iron environment in the crystal structure. Above 19 GPa, the quadrupole splittings observed in the present study are relatively constant, suggesting an increasing resistance of the lattice against further distortion.

To gain a better understanding of the connections between terrestrial planetary interiors and their surfaces, the behavior of less common minerals should be evaluated. Their interaction with the bulk materials of a planet can create meaningful deviations from an assumed average behavior. Structural and spin transitions of individual minerals or phases would likely lead to changes in the physical and chemical properties of the bulk mantle. For example, the occurrence of spin transitions in iron-bearing materials of Earth's lower mantle has received increased attention in the last 15 years owing to their potential geophysical, geochemical, and geodynamical implications [28]. At room-temperature, $(Mg,Fe)O$ ferropericlase and $FeSiO_3$ ferrosilite exhibit a similar broad spin crossover behavior with pressure [24,29], whereas the iron-bearing carbonate $FeCO_3$ siderite experiences a sharp spin transition [30]. In Figure 5, we schematically compare these transitions with szomolnokite, along with iron-bearing silicate glasses.

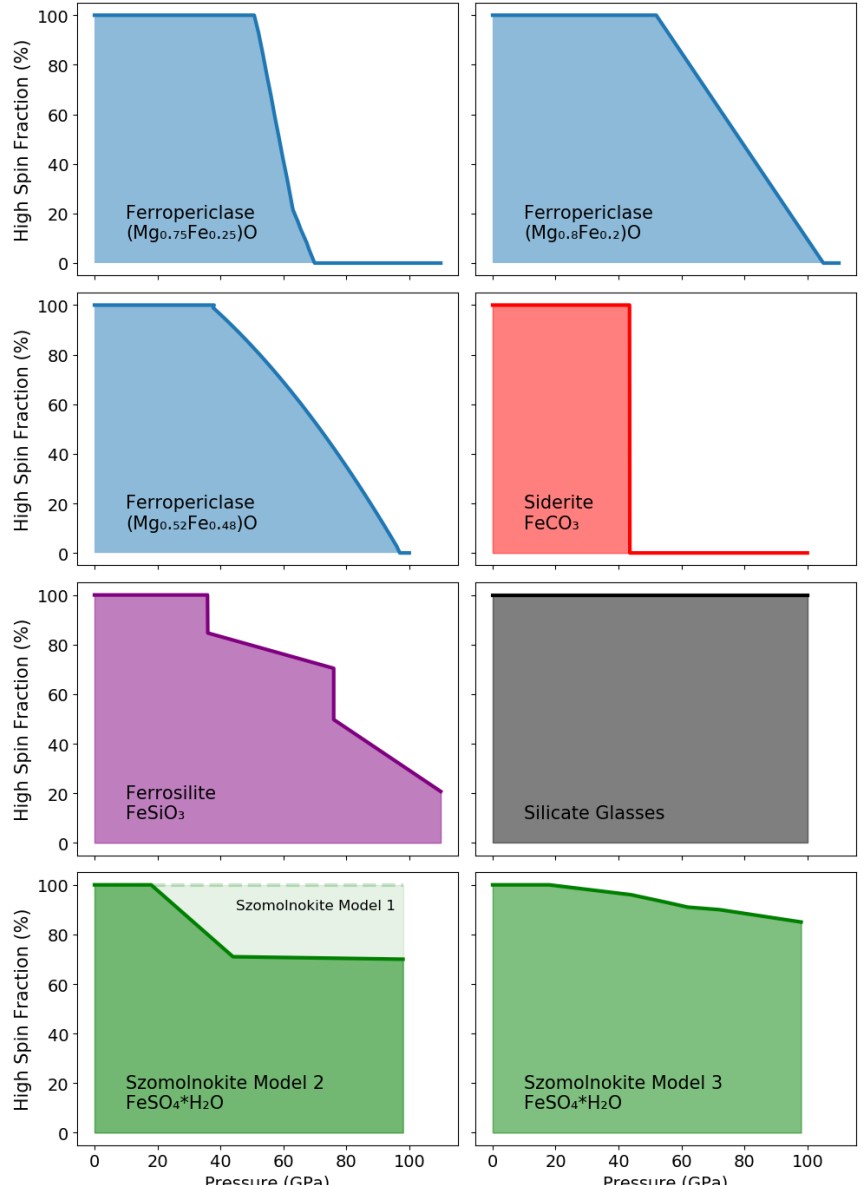

**Figure 5.** Schematic comparison of the fraction of high-spin ferrous iron (relative to low-spin ferrous iron) as a function of pressure at 300 K in various $Fe^{2+}$-bearing phases: $(Mg_{0.75}Fe_{0.25})O$ ferropericlase, [31], $(Mg_{0.8}Fe_{0.2})O$ ferropericlase [32], $(Mg_{0.52}Fe_{0.48})O$ ferropericlase [29], $FeCO_3$ siderite [30], $FeSiO_3$ ferrosilite [27], basaltic and rhyolitic glasses [24], and $FeSO_4·H_2O$ szomolnokite (this study). For szomolnokite, we include Model 2 (where site 3 undergoes a high- to low-spin transition) and Model 3 (where a fifth low-spin site gradually increases its weight fraction over a broad range of pressures). Model 1 is shown in the background of Model 2, exhibiting no spin transition over the pressure range investigated.

This study is among the first to experimentally observe the effect of pressure on the electronic environment of iron in szomolnokite, a hydrated iron sulfate. Although szomolnokite exists on the surface of Earth and likely on Mars, it is unknown if it exists at depth within either planet. Although it is also known that hydrated sulfates play important roles in the metal and hydrologic exchanges on the surface of terrestrial planets [3], we know less about the specific role that szomolnokite plays on the surfaces and interiors of planetary bodies, including icy satellites.

Our results suggest that there could be major structural transitions in szomolnokite at relatively low-pressure (Figures 3 and 4), corroborated by recent X-ray diffraction results [15]. The basalt-to-eclogite transition (>1.2 GPa, 400–1000 °C) [33] plays an important role in the context of plate tectonics

and subducting slab dynamics on Earth, owing to the large density increase across this transition [33,34]. Water content, as well as the behavior of all volatiles, can have an important effect on processes closely related to plate tectonics [35–38]. Hydrated minerals (e.g., hydrated silicates and sulfates) could react with basalt, affecting the basalt-to-eclogite phase transition, thus, influencing processes related to plate dynamics on planetary bodies. Future measurements at a range of pressure and temperature conditions are needed in order to understand the extent to which sulfate phases exist at depth in various planetary bodies and their role in volatile cycling.

**Supplementary Materials:** The following are available online at www.mdpi.com/2075-163X/10/2/146/s1, Tables S1: Lattice parameters of szomolnokite at ambient pressure and temperature conditions; Table S2: Best fit hyperfine parameters for szomolnokite according to Model 1; Table S3: Best fit hyperfine parameters for szomolnokite according to Model 2; Table S4: Best fit hyperfine parameters for szomolnokite according to Model 3; Table S5: Hydrated iron-bearing sulfates and their experimentally determined quadrupole splitting and isomer shift values at ambient pressure and temperature; Figure S1: The relative IS and FWHM of szomolnokite for Models 1, 2, and 3.

**Author Contributions:** Conceptualization, J.M.J.; methodology, G.J.F., T.P., and J.M.J.; validation, T.P., G.J.F., O.P., N.V.S., and J.M.J.; formal analysis, T.P.; investigation, T.P. and J.M.J.; writing—original draft preparation, T.P., G.J.F., N.V.S., and J.M.J.; writing—review and editing, T.P. and J.M.J.; resources, data curation, and funding acquisition, J.M.J. All authors have read and agree to the published version of the manuscript.

**Funding:** We thank the W.M. Keck Institute for Space Studies and NSF-CSEDI-EAR-1161046 for support of this research. Operations at Sector 3 (APS) and beamline 12.2.2 (ALS) are partially supported by COMPRES. This research used resources of the Advanced Photon Source and of the Advanced Light Source, which are DOE Office of Science User Facilities under contracts DE-AC02-06CH11357 and DE-AC02-05CH11231, respectively.

**Acknowledgments:** We are thankful to Wolfgang Sturhahn, Rachel Morrison, and Thomas S. Toellner for helpful discussions. We are thankful to Bob Liebermann for handling our manuscript and to three anonymous reviewers for their helpful feedback and suggestions. We thank the Mr. and Mrs. Larson for their contribution to the Summer Undergraduate Research Fellowship program at Caltech, which supported part of this work. N.V.S. was partly funded by the European Research Council (ERC) under the European Union's Horizon 2020 research and innovation program (Grant Agreement Number 681818–IMPACT).

**Conflicts of Interest:** The authors declare no conflict of interest.

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
