# Peer review of "A Synchrotron Mössbauer Spectroscopy Study of a Hydrated Iron-Sulfate at High Pressures"

_minerals, doi:10.3390/min10020146_

Round 1

Reviewer 1 Report

To the editors of Minerals,

I am writing regarding a review of the manuscript submitted to your publication by Tyler Perez and co-authors, entitled “A Synchrotron Mössbauer Spectroscopy Study of a Hydrated Iron-Sulfate at High Pressures”. The paper details the in-situ high-pressure synchrotron Mossbauer spectroscopy (SMS) of the rare (on earth) mineral Szomolnokite (FeSO4∙H2O). This is an interesting topic as SMS’s sensitivity – and selectivity - to the local iron environment is ideally suited to track both structural environments and spin states. The authors make a compelling case that, despite is relative scarcity on earth, as it was found (or speculated to exist) in other planetary bodies in the solar system, the improved knowledge Szomolnokite does have important implications in planetary geological processes. Furthermore, there are, on earth a number of other, more abundant, iron sulfate hydrates that are relevant for the geological water cycle and is therefore an improved knowledge of iron environments, especially across known pressure induced phase transitions is very relevant in the context of this chemical and/or mineral family.

While I don’t have concerns with the content or the analysis, and the papers use of English is good, I do have serious reservations regarding the structure of the presentation, as I exemplify below:

Some information given on the abstract is excessive, as since it should be a very succinct summary of the findings it typically dispenses use of references and a survey of previous work. In addition, the only mention of the structural features of Szomolnokite is given in the abstract. As a reader and especially because the models used in the involve several iron sites with different abundances, I think the structure should be presented in much greater detail. On this topic, from the text, (pg 3 lines 147-151) there is mention of a second site with only 9% of all iron content, is this borne out of the structure? Is there really such special crystallographic site that is occupied by only one in 10 irons? In fact one of the models includes a 3rd Is this crystallographic? or can be from, for instance, a sluggish phase transition? The manuscript needs to be more clear of this distinction. There is virtually no information on the synthesis route other than that is a collaboration. A little research seemed to indicate that there are straightforward routes for the synthesis, however I think for a scientific paper, at least the known route that was followed should be mentioned or referred. If a new method was used that could at least be mentioned. The authors made a good assessment that, for the audience of Minerals, it is probably necessary to explain to the readers the information gleaned from SMS and how it correlates with magneto-structural characteristics. However, despite the detailed (and a bit convoluted) discussion of the different FWHM for the different models/sites, the description of how that parameter relates to the compound (the distribution of electrical field gradients) is only mentioned on Fig 4 caption. Maybe an improvement on the clarity would be collate in a small section (a couple of paragraphs should suffice) in the introduction the description of the technique and how the fitted parameters correlate with the atomistic description of the structure. In building up models, the addition of new sites is made just to improve the fit or does have some crystal chemistry insight associated with it? The model building “philosophy” should be somewhat described. Just above the discussion section there is a very interesting observation that the quadrupole splitting seems to vary increasingly with pressure, but that this is not consistent for ambient pressure mineral series where it decreases with decreasing bond length. This is a really interesting observation and would deserved some discussion, even if just a suggestion of future work to understand the apparent contradiction. There is an interesting point on the discussion on the potential relevance of this work to the basalt to eclogite transition. Give the high level of this assertion (that is not directly related to the results) such context belongs more in the introduction as a motivation for the study than in the discussion of the results.

Besides these more structural comments, there are a few minor suggestions:

In the abstract where is “time resolved”, should be “time domain”. On line 37 reword “sulfates are of interest to planetary surfaces” On line 42 TES and MER are initials and shoul be described. On page 1 line 77-78 it mentioned that: “Hydrated sulfate minerals have very complex structures but have not yet been studied with Mössbauer spectroscopy at high pressure” however above on the same page mentions of In line 98 is Princeton Design (not Princeton Designed) In line 103 conversion to non-SI units is superfluous, and even if kept, the standard notation is psi or Kpsi. In line 147 the mention of previous Mossbauer studies is probably laboratory-based measurements (as earlier was mentioned this is the first SMS measurements on this mineral). Either way this ambiguity should be clarified. On line 172, replace “ (…) large amounts of fluctuation” by large fluctuations. On that topic, fig 4 does not seem to agree with this characterization. On line 174-175, the mention: ” FWHM was kept constant up to” means "was constrained" or "was observed to be"? Clarify. On caption of fig 4 Determined using CONUSS is redundant. I suggest removing it. Figure 5 can easily be condensed into a single plot and all panels share units, scales and consist of a single line

In summary, once the paper is structured in a more clear way, I find this work suitable for publication in Minerals.

Best regards   

Reviewer 2 Report

The work by Perez et al. reports on the high-pressure Mössbauer study of szomolnokite. According to the authors, a pressure-induced spin transition is suggested to initiate above 18 GPa.

I think the manuscript is of interest to the readership of the Minerals journal, but I would recommend the following additions:

1) Even though the abstract describes the structure of szomolnokite, there is nothing relevant in the manuscript (Figure or description). I think it would benefit the readers if the authors included a description of the structure, which would facilitate in turn the "visualization" of the spin transition process 

2) The Mössbauer spectra spectra fitting appear a little bit biased to my understanding, in the sense that you can always add more components, improve the fitting merit number, and justify their use and physical meaning whatsoever. Before proposing a spin transition (which to me appears to say the least unclear), I would like the authors to show the high-spin only model (mentioned only in passing) as a separate Figure. And then of course compare the various models, but referring to Figures.

Upon these minor revisions, I would me more than happy to look at the manuscript again.

Reviewer 3 Report

This article describes a detailed pressure dependence of szomolnokite by synchrotron radiation Mössbauer spectroscopy. Szomolnokite was sometimes seen in minerals and thus its temperature and pressure dependence is important to the earth and planetary science. In fact, it might be important to understand the inner Venus, where vast amount of sulphide is considered to be. There is no doubt that the high pressure experiments of minerals are critical method to understand the core of planets. Thus, there is also no doubt about the importance of this article. However, the analysis of the time spectra of nuclear resonant forward scattering (NFS) does not look sufficiently reasonable to me. Considering these importance and the status of analysis, I recommend publication after a major revision. The following is my questions and comments to improve this manuscript.

Major points

 First, I do not know the detailed algorithm applied in CONUSS. Nevertheless, I think the results shown in tables S2 and S3, and figures 4 and S1 includes too many information compared to the apparently simple time spectra shown in figure 3. I can only see four quantum beats in the spectra below 11.6 GPa, although authors assume three components which includes at least three doublet Mössbauer spectra (that is, at least six nuclear resonance). I think some other assumptions are required to obtain the results. Similarly, I can only see three or four modulated quantum beat patterns, although authors assume four or five components. The modulation might be owing to the additional iron site(s), as assumed by authors, but also might be owing to the combination of quantum and dynamical beats. Please describe the additional assumption to analyze these time spectra, including the reason why some parameters are fixed and its reason, at least supplementary materials.

 Second, I do not agree with the fixed Lamb-Mössbauer factor, because this factor is affected by pressure. For example, Prof. Lübbers clearly showed the enhanced Lamb-Mössbauer factor of alpha-iron by nuclear resonant inelastic scattering (Science, vol. 287, page 1250 (2000)). Please consider the effect of high pressure to Lamb-Mössbauer factor. The time spectra of NFS is very sensitive method and thus the pressure dependence sometimes drastically changes their apparent shape.

Third, Prof. Lübbers also showed the elevated Debye temperature, that results in the change of second order Doppler shifts. Therefore, the experimental apparent isomer shifts of site 1 written in tables S2 and S3 would not be 1.23 mm/s in all pressure range. I suggest that these parameters should be written as delta IS, as the authors used in figure S1.

 Fourth, I would like to ask the authors if the site 2 in ambient pressure is really FeSO4_H2O. This chemical compound has only one crystallographic iron site and thus its Mössbauer spectra should show one site. In fact, the XRD pattern in figure 1 showed no additional site. Is there really second site in the time spectrum at ambient pressure? Otherwise, is there really no additional chemical compounds contaminated in the period among the XRD and SRMS?

 Fifth, I would like to also ask the authors if the FWHM really mean the distribution of QS. Strictly speaking, the FWHM should be the natural linewidth of Fe-57, 0.097 mm/s in the calculation of time spectra of NFS and the distribution of hyperfine parameters should be considered another way. In the analysis of many Mössbauer absorption spectra, FWHM is used to approximately express the distribution, but it should not be in the analysis of the time spectra, because the distribution of the hyperfine parameters modulates the coherence of the resonant path in another way where simple broadening of FWHM affects. If you have some trouble in the fitting of the spectra, the consideration of this effect might be helpful.

Minor points

1, The usage of the word “FWHM” is not common. This word usually means the width of Mössbauer absorption spectra and not the distribution of quadrupole splitting parameters. Many Mössbauer scientists feel confused about the “FWHM” less than the natural linewidth of Fe-57. Please use more detailed description or other abbreviation, such as “W_QS”.

2, Line 8: John Hopkins University is very famous as the California Institute of Technology be. Therefore, if you do not abbreviate Caltech, you should not abbreviate John Hopkins University, too.

3, Line 43: What does “MER” mean? Mar explorer rover? Please write the full form before the first description of abbreviation.

4, Line 50: “which” should be “who”, because the antecedent is “Fortes et al.”.

5, Line 73: synchrotron Mössbauer spectroscopy, abbreviated as “SMS” in line 106 is described here. Please use the abbreviation in the first description. Furthermore, I recommend the abbreviation of “SRMS” for synchrotron radiation Mössbauer spectroscopy because SMS is also the abbreviation of “synchrotron Mössbauer source”, first constructed by Dr. Chumakov using nuclear Bragg monochromator developed by Prof. Smirnov.

7, Line 103: “tor” should be “torr”. Moreover, I recommend that torr and kbars in this line is rewritten in the unit of Pa, although these units are still written in many gauges.

8, Line 277: “Mvssbauer” should be “Mössbauer”. In addition, please add the title of the journal, for example, Lunar and Planetary Science XXXVI, 2005, abstract No. 2108.

Round 2

Reviewer 2 Report

The authors have revised their work appropriately and answered all of my remarks. Therefore, I recommend publication of their work.

Author Response

We thank the reviewer for the insightful feedback that has helped us improve the manuscript.